# ON THE POWER OF GRAPH NEURAL NETWORKS AND THE ROLE OF THE ACTIVATION FUNCTION

## ABSTRACT

In this article we present new results about the expressivity of Graph Neural Networks (GNNs). We prove that for any GNN with piecewise polynomial activations, whose architecture size does not grow with the graph input sizes, there exists a pair of non-isomorphic rooted trees of depth two such that the GNN cannot distinguish their root vertex up to an arbitrary number of iterations. The proof relies on tools from the algebra of symmetric polynomials. In contrast, it was already known that unbounded GNNs (those whose size is allowed to change with the graph sizes) with piecewise polynomial activations can distinguish these vertices in only two iterations. Our results imply a strict separation between bounded and unbounded size GNNs, answering an open question formulated by Grohe (2021). We next prove that if one allows activations that are not piecewise polynomial, then in two iterations a single neuron perceptron can distinguish the root vertices of any pair of nonisomorphic trees of depth two (our results hold for activations like the sigmoid, hyperbolic tan and others). This shows how the power of graph neural networks can change drastically if one changes the activation function of the neural networks. The proof of this result utilizes the Lindemann-Weierstrauss theorem from transcendental number theory.

## 1 INTRODUCTION

Graph Neural Networks (GNNs) form a popular framework for a variety of computational tasks involving network data, with applications ranging from analysis of social networks, structure and functionality of molecules in chemistry and biological applications, computational linguistics, simulations of physical systems, techniques to enhance optimization algorithms, to name a few. The interested reader can look at Zitnik et al. (2018); Bronstein et al. (2017); Ding et al. (2019); Defferrard et al. (2016); Scarselli et al. (2008); Hamilton (2020); Duvenaud et al. (2015); Khalil et al. (2017); Stokes et al. (2020); Zhou et al. (2020); Battaglia et al. (2016); Cappart et al. (2021); Sanchez-Gonzalez et al. (2020), which is a small sample of a large and actively growing body of work.

Given the rise in importance of inference and learning problems involving graphs and the use of GNNs for these tasks, significant progress has been made in recent years to understand their computational capabilities. See the excellent recent survey Jegelka (2022) for an exposition of some aspects of this research. One direction of investigation is on their so-called *separation power* which is the ability of GNNs to distinguish graphs with different structures. In this context, it becomes natural to compare their separation power to other standard computation models on graphs, such as different variants of the *Wesfeiler-Lehman algorithm* Cai et al. (1992); Xu et al. (2018); Huang & Villar (2021), and the very closely related *color refinement algorithm* Grohe (2021). These investigations are naturally connected with *descriptive complexity theory*, especially to characterizations in terms of certain logics; see Grohe (2017; 2021) for excellent introductions to these different connections. A closely related line of work is to investigate how well general functions on the space of graphs can be approximated using functions represented by GNNs; see Azizian & Lelarge (2021); Keriven & Peyré (2019); Maron et al. (2019); Geerts & Reutter (2022) for a sample of work along these lines. Our work in this paper focuses on improving our understanding of the separation power of GNNs.

At a high level, the computational models of GNNs, Wesfeiler-Lehman/color refinement type algorithms and certain logics in descriptive complexity are intimately connected because they all fall under the paradigm of trying to discern something about the global structure of a graph from local

neighborhood computations. Informally, these algorithms iteratively maintain a state (a.k.a. "color") for each vertex of the graph and in every iteration, the state of a vertex is updated by performing some predetermined set of operations on the set of current states of its neighbors (including itself). The different kinds of allowed states and allowed operations determine the computational paradigm. For instance, in GNNs, the states are typically vectors in some Euclidean space and the operations for updating the state are functions that can be represented by deep neural networks. As another example, in the color refinement algorithm, the states are multisets of some predetermined finite class of labels and the operations are set operations on these multisets. A natural question then arises: Given two of these models, which one is more powerful, or equivalently, can one of the models always simulate the other? Several mathematically precise answers to such questions have already been obtained. For instance, it has been proved independently by Morris et al. (2019) and Xu et al. (2018) that the color refinement algorithm precisely captures the expressiveness of GNNs in the sense that there is a GNN distinguishing two nodes of a graph (by assigning them different state vectors) if and only if color refinement assigns different multisets to these nodes. Such a characterization holds for *unbounded* GNNs, i.e. GNNs for which the underlying neural network sizes can grow with the size of the input graphs. This implies a characterisation of the distinguishability of nodes by GNNs as being equivalent to what is known as *Graded Modal Counting Logic* (GC2); see Barceló et al. (2020) for some recent, quantitatively precise results in this direction.

Reviewing these equivalences in a recent survey Grohe (2021), Grohe emphasizes the fact that the above mentioned equivalence between the separation power of GNNs and the color refinement algorithm has only been established for unbounded GNNs whose neural network sizes are allowed to grow as a function of the size of the input graphs. Question 1 on his list of important open questions in this topic asks what happens if one considers *bounded* GNNs, i.e., the the size of the neural networks is fixed a priori and cannot change as a function of the size of the input graphs. *Do bounded GNNs have the same separation power as unbounded GNNs and color refinement?* We answer this question in the negative, by constructing, for any given bounded GNN, two non isomorphic rooted trees of depth two such that their root nodes cannot be distinguished by the GNN. Interestingly, only the sizes of the trees depend on the GNN, but their depth does not. This result is stated formally in Theorem 3 and it holds for bounded GNNs with piecewise polynomial activations (this includes, e.g., ReLU activations). We prove a second result that shows how the activation function dramatically impacts the expressivity of bounded size GNNs: if one allows activation functions that are not piecewise polynomial, all root nodes of rooted trees of depth two can be distinguished by a single neuron perceptron. This result is formally stated in Theorem 4.

The rest of this article is organized as follows. In Section 2 we present the main definitions and formal statement of our results. In Section 3 we give an overview of the proofs. Sections 4 and 5 fill in the technical details.

## 2 FORMAL STATEMENT OF RESULTS

We assume graphs to be finite, undirected, simple, and vertex-labelled: a graph is a tuple $G = (V(G), E(G), P_1(G), ..., P_\ell(G))$ consisting of a finite vertex set $V(G)$, a binary edge relation $E(G) \subset V(G)^2$ that is symmetric and irreflexive, and unary relations $P_1(G), \cdots, P_\ell(G) \subset V(G)$ representing $\ell > 0$ vertex labels. In the following, the number $\ell$ of labels, which we will also call *colors*, is supposed to be fixed and does not grow with the size of the input graphs. When there is no ambiguity about which graph $G$ is being considered, $N(v)$ refers to the set of neighbors of $v$ in $G$ not including $v$. $|G|$ will denote the number of vertices of $G$. We use simple curly brackets for a set $X = \{x \in X\}$ and double curly brackets for a multiset $Y = \{\{y \in Y\}\}$. For a set $X$, $|X|$ is the cardinal of $X$. When $m$ is a positive integer, $\mathfrak{S}_m$ is the set of permutations of $\{1, \cdots, m\}$.

**Definition 1** (Piecewise polynomial). *Let $m$ be a positive integer. A function $f : \mathbb{R}^m \to \mathbb{R}$ is piecewise polynomial iff there exist multivariate polynomials $P_1, \cdots, P_r \in \mathbb{R}[X_1, \cdots, X_m]$ such that for any $x \in \mathbb{R}^m$, there exists $i \in \{1, \cdots, r\}$ such that $f(x) = P_i(x)$. The degree of a piecewise polynomial function $f$ is $\deg(f) := \max\{\deg(P_1), \cdots, \deg(P_r)\}$. The number of polynomial pieces of a piecewise polynomial $f$ is the smallest $r$ such that $f$ can be represented as above.*

**Definition 2** ((Elementary) Symmetric polynomial). *For any positive integer $m$, a polynomial $P \in \mathbb{R}[X_1, \cdots, X_m]$ is said to be symmetric if for any permutation $\pi \in \mathfrak{S}_m$ of $\{1, \cdots, m\}$ and any $v_1, \ldots, v_m \in \mathbb{R}$, $P(v_{\pi(1)}, \cdots, v_{\pi(m)}) = P(v_1, \cdots, v_m)$. For any $k \in \{1, \cdots, m\}$, the elementary symmetric polynomial $s_k$ is given by $s_k(X_1, \cdots, X_m) := \sum_{1 \le j_1 < j_2 < \cdots < j_k \le m} X_{j_1} \cdots X_{j_k}$.*

**Definition 3** (Embedding, equivariance, and refinement). *Given a set $X$, an embedding $\xi$ is a function that takes as input a graph $G$ and a vertex $v \in V(G)$, and returns an element $\xi(G, v) \in X$ for each vertex $v$ of the graph. An embedding is* equivariant *if and only if for any pair of isomorphic graphs $G, G'$, and any isomorphism $f$ from $G$ to $G'$, it holds that $\xi(G, v) = \xi(G', f(v))$. We say that $\xi$ refines $\xi'$ if and only if for any graph $G$ and any $v \in V(G), \xi(G, v) = \xi(G, v') \implies \xi'(G, v) = \xi'(G, v')$.*

**Definition 4** (Color refinement). *Given a graph $G$, and $v \in V(G)$, let $(G, v) \mapsto \mathsf{col}(\mathsf{G}, \mathsf{v})$ be the function which returns the color of the node $v$. The color refinement refers to a procedure that returns a sequence of equivariant embeddings $cr^t$, computed recursively as follows:*

- $cr^0(G, v) = \mathsf{col}(G, v)$

- *For $t \geq 0$, $\mathsf{cr}^{t+1}(G, v) := (\mathsf{cr}^t(G, v), \{\{\mathsf{cr}^t(G, w) : w \in N(v)\}\})$*

*In each round, the algorithm computes a coloring that is finer than the one computed in the previous round, that is, $\mathsf{cr}^{t+1}$ refines $\mathsf{cr}^t$. For some $t \leq n := |G|$, this procedure stabilises: the coloring does not become strictly finer anymore.*

**Remark 1.** *We refer the reader to the seminal work (Cai et al., 1992, Sections 2 and 5) for comments about the history and connections between the color refinement and Weisfeiler-Lehman algorithms.*

**Definition 5** (Graph Neural Network (GNN)). *A GNN is a recursive embedding of vertices of a labelled graph by relying on the underlying adjacency information and node features. Each vertex $v$ is attributed an indicator vector $\xi^0(v)$ of size $\ell$, encoding the color of the node $v$: the colors being indexed by the palette $\{1, \cdots, \ell\}$, $\xi^0(v) = e_i$ (the $i$-th canonical vector) if the color of the vertex $v$ is $i$. The GNN is fully characterized by:*

$\circ$ *A combination function $\mathsf{comb} : \mathbb{R}^{2\ell} \longrightarrow \mathbb{R}^\ell$ which is a feedforward neural network with given activation function $\sigma : \mathbb{R} \longrightarrow \mathbb{R}$.*

$\circ$ *The update rule of the GNN at iteration $t \in \mathbb{N}$ for any labelled graph $G$ and vertex $v \in V(G)$, is given as follows:*

$$\xi^0(v) \text{ is the indicator vector of the color of } v, \qquad \xi^{t+1}(v) = \mathsf{comb}(\xi^t(v), \sum_{w \in N(v)} \xi^t(w))$$

**Remark 2.** *This type of GNN is sometimes referred to as a **recurrent GNN**. The general definition (cf. for instance Grohe (2021)) usually considers a sequence of combine and aggregation functions which may depend on the iteration $t$. The aggregation functions replaces the sum over the neighborhood, i.e. at each iteration $\mathsf{comb}(\xi^t(v), \mathsf{agg}(\{\{\xi^t(w) : w \in N(v)\}\}))$ is applied. It has been proved in Xu et al. (2018) that for any $\mathsf{agg}$ function, there is a GNN (of potentially larger size) whose aggregation function is the summation and which refines any GNN with this aggregation function. The results of this article extend to GNNs whose combination and aggregation functions are allowed to be different in different iterations, but are multivariate piecewise polynomials. For ease of presentation, we restrict to recurrent GNNs.*

Given these definitions, we can now formally state the previously known results about the expressivity of unbounded GNNs (Theorems 1 and 2). Namely, in Theorem 2, the size of the GNN is allowed to grow with $n$.

**Theorem 1.** *Grohe (2021); Xu et al. (2018); Morris et al. (2019) Let $d \geq 1$, and let $\xi^d$ be a vertex invariant computed by a GNN after $d$ iterations. Then $\mathsf{cr}^d$ refines $\xi$, that is, for all graphs $G, G'$ and vertices $v \in V(G), v' \in V(G')$, $\mathsf{cr}^d(v) = \mathsf{cr}^d(v') \implies \xi^d(G, v) = \xi^d(G', v')$.*

**Theorem 2.** *(Grohe, 2021, Theorem VIII.4)Xu et al. (2018); Morris et al. (2019) Let $n \in \mathbb{N}$. Then there is a recurrent GNN such that for all $t = 0, \cdots, n$, the vertex invariant $\xi^t$ computed in the $t$-th iteration of the GNN refines $\mathsf{cr}^t$ on all graphs of order at most $n$.*

In contrast, we prove Theorems 3 and 4 for bounded GNNs:

**Theorem 3.** *For any GNN, i.e., choice of combination function, represented by a feedforward neural network with piecewise polynomial activation, and any natural number $I \in \mathbb{N}$, there exists a pair of rooted trees $T$ and $T'$ (unicolored, i.e. $\ell = 1$) of depth two with root nodes $s$ and $s'$ respectively such that:*

- $\mathsf{cr}^2(T, s) \neq \mathsf{cr}^2(T', s')$, *i.e. $s$ and $s'$ can be distinguished with color refinement in two iterations.*

- $\xi^t(T, s) = \xi^t(T', s')$ for all $t \leq I$, *i.e $s$ and $s'$ cannot be distinguished by the GNN until iteration $I + 1$.*

**Theorem 4.** *In two iterations, a single neuron perceptron with an activation that is not piecewise polynomial such as $\sigma \in \{\exp, \text{sigmoid}, \cosh, \sinh, \tanh\}$ can distinguish the root nodes of any pair of non-isomorphic rooted trees of depth two.*

**Remark 3.** *Furthermore, our proof also shows that for a GNN to distinguish the source vertices of any pair of trees of depth $2$ (displayed in Figure 1) with $m$ vertices at depth one and with prescribed degrees $k_1, \cdots, k_m \in \{1, \cdots M\}$, it is necessary that the number of regions $r$ of the underlying neural network of degree $q$ verifies:*

$$r \geq \binom{M + m + 1}{m} \frac{1}{M^{q^2}(\binom{m}{q}q)^q} \tag{1}$$

*This condition may not be sufficient, as the polynomial regions have also to be correctly placed. In particular, for ReLU neural networks ($q = 1$), there exist upper bounds for the number of regions given the number of layers and neurons per each layer (cf. Montufar et al. (2014); Montúfar (2017) and Serra et al. (2018) for refinements). Therefore, for a ReLU neural network with $L$ layers, input dimension $n_0$ and with $n_1, \cdots, n_L$ neurons per layer, it is necessary that*

$$\prod_{l=1}^{L-1} \left\lfloor \frac{n_l}{n_0} \right\rfloor^{n_0} \cdot \sum_{j=0}^{n_0} \binom{n_L}{j} \geq \binom{M + m + 1}{m} \frac{1}{M \cdot m}$$

*One could also derive similar lower bounds in the general piecewise polynomial case if provided an upper bound on the number of polynomial regions given the number of layers and neurons per layer.*

## 3 OVERVIEW OF THE PROOFS

To establish our first result, we will use rooted trees of the form shown in Figure 1 which is a tree of depth two whose depth one vertices have prescribed degrees $k_1, \ldots, k_m$, with $k_1, \ldots, k_m \geq 1$. Given a GNN with piecewise polynomial activation and a natural number $I \in \mathbb{N}$, we will show that there exist two sets of integers $k_1, \cdots, k_m$ and $k'_1, \cdots, k'_m$ that are not the same up to permutations, such that for the corresponding rooted trees $T[k_1, \cdots, k_m]$ and $T[k'_1, \cdots, k'_m]$, the GNN cannot distinguish $s$ and $s'$ for the first $I$ iterations, i.e. $\xi^t(T, s) = \xi^t(T', s')$ for any $t \in \{1, \cdots, I\}$. Note that the natural numbers $m$, and $k_1, \cdots, k_m$ and $k'_1, \cdots, k'_m$ will depend on $I$, the activation and the size of the neural network considered.

The proof of the first result is structured as follows. Since the trees are parameterized by $m$-tuples of integers $k_1, \ldots, k_m$, the embedding of the root node computed by the GNN at any iteration is a function of these $m$ integers. Since the activations are piecewise polynomials, these embeddings of the root node are also piecewise multivariate symmetric polynomial functions of $k_1, \ldots, k_m$ (Lemma 3). Then, we show that there exists a large enough region of $\mathbb{R}^m$ on which this piecewise polynomial function is evaluated by the *same* polynomial. This region is large enough in the following sense: we prove that it contains more integral vectors than the number of possible values a symmetric polynomial with bounded degree can take on these vectors, even after identifying vectors up to permutations of the coordinates. This implies that the polynomial will take the same value on two distinct integral vectors whose coordinates are not identical up to permutations. When translating this result back to the world of GNNs, this implies that the two embeddings of the root nodes of the trees corresponding to these two vectors will coincide. To conclude a separation between bounded and unbounded GNNs, we justify that the unbounded ones can seperate these two vertices. This is based on the previous result (Theorem 2) stating that unbounded GNNs refine color refinement.

Our second result states that for activations that are not piecewise polynomial, a one neuron perceptron GNN can distinguish the root nodes of any pair of nonisomorphic trees of depth two. In particular, we prove this when the activation function is the exponential, the sigmoid or the hyperbolic sine, cosine or tangent functions. This is done by showing that the condition $\xi^2(s) = \xi^2(s)$ corresponds to a relation between the exponentials of the integers $k_1, \cdots, k_m$ and $k'_1, \cdots, k'_m$. Applying the Lindemann-Weirstrass Theorem in transcendental number theory (Lemma 4 and Theorem 5) leads to the conclusion that $k'_1, \ldots, k'_m$ must be a permutation of $k_1, \ldots, k_m$, showing that the trees are isomorphic.

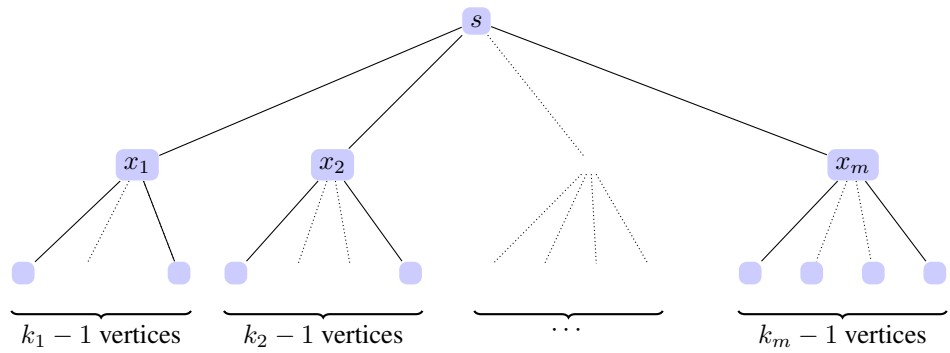

Figure 1: $T[k_1, \cdots, k_m]$

## 4 COLLISION WITH PIECEWISE POLYNOMIAL ACTIVATIONS

The following statement is a reformulation of the fundamental theorem of symmetric polynomials, where we added a constraint on the degree of the decomposition. We provide a proof for completeness.

**Lemma 1.** *Let $P$ be a symmetric multivariate polynomial of $m$ variables of degree $q$ with $q \leq m$. Then $P$ can be written as a polynomial of degree $q$ of the elementary symmetric polynomials $s_1, \cdots, s_q$.*

*Proof.* For $\alpha = (\alpha_1, \cdots, \alpha_m) \in \mathbb{N}^m$, define the multidegree of a monomial $\mathsf{mdeg}(X_1^{\alpha_1} \cdots X_m^{\alpha_m}) := \sum_{i=1}^{m} \alpha_i (q+1)^{m-i}$. By definition, the *leading term* of a polynomial is the monomial with greatest multidegree. We present a proof by induction on the multidegree of $P$. We first need the following claim.

*Claim: Let $P \in \mathbb{R}[X_1, \cdots, X_m]$ be a symmetric polynomial. Let $c_\alpha X^\alpha = c_\alpha X_1^{\alpha_1} \cdots X_m^{\alpha_m}$ be the leading term of $P$ ($c_\alpha \neq 0$). Then $c_\alpha X^{\pi(\alpha)}$ is also a monomial in $P$, for every permutation $\pi$ of $\{1, \cdots, m\}$.*

*Proof of the claim.* Define $\phi^\pi : \mathbb{R}[X_1, \cdots, X_m] \rightarrow \mathbb{R}[X_1, \cdots X_m]$, be the linear map between polynomials which maps variable $X_i$ to $X_{\pi(i)}$. Since $P$ is symmetric, then $\phi^\pi(P) = P$ (we use here the equivalence between equality of $\phi^\pi(P)$ and $P$ as functions and as formal algebraic objects). Since $c_\alpha X^\alpha$ is a term in $P$ and $c_\alpha X^{\pi(\alpha)}$ is a monomial in $\phi^\pi(P)$ for any permutation $\pi$, we must have $c_\alpha X^{\pi(\alpha)}$ as a monomial in $P$. □

*Base case.* The property is true for any polynomial of multidegree $0$ (constant polynomial).

*Induction step.* Let $c_\alpha X^\alpha$ be the leading term of $P$. Then $c_\alpha X^{\pi(\alpha)}$ is a monomial in $P$ for every $\pi \in \mathfrak{S}_m$ by the previous claim. Therefore, the leading term's exponent $\alpha = (\alpha_1, \cdots, \alpha_m)$ must satisfy $\alpha_1 \geq \alpha_2 \geq \cdots \geq \alpha_m$. Since $P$ has degree $q$ then $\alpha_i = 0$ for any $i \geq q+1$.

Let $d_m = \alpha_m, d_{m-1} = \alpha_{m-1} - \alpha_m, \cdots, d_i = \alpha_i - \alpha_{i+1}, \cdots, d_1 = \alpha_1 - \alpha_2$. Define $Q'(X_1, \cdots, X_m) := s_1^{d_1} \cdots s_m^{d_m}$. $Q'$ is a polynomial of $s_1, \cdots, s_q$ and the leading term of $Q'$ is $c'_\alpha X^\alpha$ where $c'_\alpha \neq 0$. In particular, $\deg(Q') \leq q$.

Now, let $P' := P - \frac{c_\alpha}{c'_\alpha} Q'$. Then $\mathsf{mdeg}(P') < \mathsf{mdeg}(P)$, and $P'$ is symmetric because $P$ and $Q'$ are. Applying the induction hypothesis to $P'$, get $Q''$ such that $P' = Q''(s_1, \cdots, s_q)$. Define $Q := \frac{c_\alpha}{c'_\alpha} Q' + Q''$. $Q$ is a polynomial of degree at most $q$ of $s_1, \cdots, s_q$ and $Q = P$, which completes the induction step. □

**Lemma 2.** *Let $q$ be a positive integer. Then, there exists $m \in \mathbb{N}$ and two integral vectors $(k_1, \cdots, k_m) \in \mathbb{N}^m$ and $(k'_1, \cdots, k'_m) \in \mathbb{N}^m$ that are not equal up to a permutation such that for any sequence of symmetric piecewise polynomial functions $(f_p : \mathbb{R}^p \rightarrow \mathbb{R})_{p \in \mathbb{N}}$ satisfying $\deg(f_p) \leq q$ for all $p \in \mathbb{N}$ (bounded degree condition on each polynomial piece for any $p$), $f_m(k_1, \cdots, k_m) = f_m(k'_1, \cdots, k'_m)$.*

*Proof.* Let $m$ be any natural number such that $m > \max\{q^2, 2q\}$. For any natural number $M$, let $F_M$ be the box $\{(k_1, \cdots, k_m) \in \mathbb{Z}^m : \forall i \quad 1 \le k_i \le M\}$. Let $\Omega_M := \{\{\{x_1, \cdots, x_m\}\} : (x_1, \cdots, x_m) \in F_M\}$; in other words, $\Omega_M$ is the set of multisets of size $m$ whose elements, when arranged as vectors, are in the box $F_M$[1]. Consider

$$\Phi : \quad \Omega_M \longrightarrow \mathbb{Z}^q$$
$$S \mapsto (s_1(S), s_2(S), \cdots, s_q(S))$$

which is well-defined because of the symmetry of the elementary symmetric polynomials $s_1, \ldots, s_q$. Note that for any $i = 1, \ldots, q$, $s_i$ is a sum of $\binom{m}{i}$ monomials whose maximum value is $M^i$ on $F_M$. Therefore, $|\mathsf{Im}(\Phi)| \le (\sum_{i=1}^q \binom{m}{i} M^i)^q \le M^{q^2}(\binom{m}{q} q)^q$ where the last inequality follows from $\binom{m}{i} \le \binom{m}{q}$ because $m > 2q$.

On the other hand, $|\mathsf{Dom}(\Phi)| = \binom{M+m-1}{m}$ (number of multisets of size $m$ whose elements are taken from a set of size $M$). Now, let $f_m : \mathbb{R}^m \to \mathbb{R}$ be the $m$-th function in the given sequence of symmetric piecewise polynomial functions where each polynomial piece has degree at most $q$. Let $r$ be the number of pieces of $f_m$. Then, there is a subset of $\mathsf{Dom}(\Phi)$ with at least $\frac{1}{r}\binom{M+m-1}{m}$ elements where $f_m$ is a symmetric polynomial $P$ of degree at most $q$. Lemma 1 tells us that $P$ can be expressed as a polynomial of degree at most $q$ of the elementary symmetric polynomials $s_1, \cdots, s_q$. Due to the pigeonhole principle, any such polynomial will be equal on two distinct multisets $\{\{k_1, \cdots, k_m\}\}$ and $\{\{k'_1, \cdots, k'_m\}\}$ in $\Omega_M$ as soon as:

$$\underbrace{\frac{1}{r}\binom{M+m-1}{m}}_{\text{number of points}} \quad > \quad M^{q^2}(\binom{m}{q} q)^q \quad \ge \quad \underbrace{|\mathsf{Im}(\Phi)|}_{\text{number of values } P \text{ can take at most}} \quad (2)$$

Such a value for $M$ can be found by noticing that $\binom{M+m-1}{m}$ is a polynomial of $M$ of degree $m$ whereas $M^{q^2}(\binom{m}{q} q)^q$ is a polynomial of $M$ of degree $q^2$. Since we chose $m$ to be greater than $q^2$, there exists $M \in \mathbb{N}$ such that Equation 2 is true. Hence there exist $k$ and $k'$ whose coordinates are not equal up any permutation and such that $s_i(k_1, \cdots, k_m) = s_i(k'_1, \cdots, k'_m)$ for any $i \in \{1, \cdots, m\}$. In turn $f_m(k_1, \cdots, k_m) = f_m(k'_1, \cdots, k'_m)$.

$\square$

**Lemma 3.** *Let $\xi^t(T[k_1, \ldots, k_m], s)$ be the embedding obtained via a GNN with piecewise activation functions after $t$ iterations, where $\xi^0(v) = 1$ for all vertices $v \in V(T[k_1, \ldots, k_m])$. Then, for any iteration $t$, there exists a symmetric multivariate piecewise polynomial function $F$ such that $\xi^t(T[k_1, \ldots, k_m], s) = F(k_1, \cdots, k_m)$. Furthermore, the degree of $F$ does not depend on $m$, but only on the underlying neural network and $t$.*

*Proof.* We first prove by induction on $t$ that, for any vertex $v \in V(T[k_1, \cdots, k_m])$, $\xi^t(T[k_1, \ldots, k_m], v)$ is a piecewise polynomial function of the $k_i$'s.

Base case: for $t = 0$ this is trivial since all vertices are initialised with the constant polynomial 1, whose degree does not depend on $m$.

Induction step: Suppose the property is true at iteration $t$, i.e for each node $w$, $\xi^t(T[k_1, \ldots, k_m], w)$ is a multivariate polynomial of the $k_i$'s. Since

$$\xi^{t+1}(T[k_1, \ldots, k_m], v) = \phi(\xi^t(T[k_1, \ldots, k_m], v), \sum_{w \in N(v)} \xi^t(T[k_1, \ldots, k_m], w))$$

where $\phi$ is a piecewise multivariate polynomial of $k_1, \cdots, k_m$, by composition $\xi^{t+1}(T[k_1, \ldots, k_m], v)$ is a multivariate polynomial of $k_1, \cdots, k_m$. By induction the degree of $\xi^{t+1}(T[k_1, \ldots, k_m], v)$ depends only on $t$ and the degree of $\phi$, which does not depend on $m$.

Finally, we know from (Grohe, 2021, Theorem VIII.1) that the color refinement algorithm refines any GNN at any iteration. Since the tuple obtained by color refinement for the vertex $s$ is invariant with respect to permutations of the $k_i$'s, $\xi^t(T[k_1, \ldots, k_m], s)$ is also invariant with respect to permutations of the $k_i$'s.

$\square$

---

[1]Another way to define $\Omega_M$ is as the orbits of the action of the symmetric group $\mathfrak{S}_m$ on $F_M$.

*Proof of Theorem 3.* We already know Grohe (2021) that color refinement refines any recurrent GNN (even with an architecture of unbounded size). We prove the existence of pairs of graphs that can be separated by the color refinement algorithm, but cannot be separated by a recurrent GNN of fixed (but arbitrary) size. We use $T[k_1, \cdots, k_m]$ to refer to the tree illustrated in Figure 1. This tree has depth two, a root node $s$, and contains $m$ nodes at depth one. Each vertex $i$ at depth 1 has exactly $k_i - 1$ "children" at depth two (and therefore $k_i$ neighbors, where $k_i$ is a positive integer). In the following, all vertices have color label 1.

*Claim: Let $T[k_1, \cdots, k_m]$ and $T'[k'_1, \cdots, k'_m]$ be two rooted trees given by Figure 1. If the $k_i$'s and $k'_i$'s are not equal up to a permutation, the color refinement distinguishes $s$ and $s'$ after two iterations, i.e. $\mathsf{cr}^2(s) \neq \mathsf{cr}^2(s')$.*

*Proof of claim.* Simply note that

$$\mathsf{cr}^2(s) = (\mathsf{cr}^1(s), \{\{\mathsf{cr}^1(x_1), \cdots, \mathsf{cr}^1(x_m)\}\})$$

$$\mathsf{cr}^1(s) = (\underbrace{1}_{\mathsf{cr}^0(s)}, \{\{\underbrace{1, \cdots, 1}_{m \text{ times}}\}\})$$

$$\forall i \in \{1, \cdots, m\} \quad \mathsf{cr}^1(x_i) = (\underbrace{1}_{\mathsf{cr}^0(x_i)}, \{\{\underbrace{1, \cdots, 1}_{k_i \text{ times}}\}\})$$

hence $\mathsf{cr}^2(s)$ is uniquely determined by the multiset $\{\{k_1, \cdots, k_m\}\}$. $\quad\square$

Let $T > 0$ be a positive integer, and for $0 \leq t \leq T$, let $f_t(k_1, \cdots, k_m) := \xi^t(T[k_1, \ldots, k_m], s)$ be the value returned by a GNN with piecewise polynomial activation after $t$ iterations (note that the embeddings are one-dimensional because only one color is used). Due to Lemma 3, for any $t \in \{0, \cdots, T\}$, $((k_1, \cdots, k_m) \mapsto f_t(k_1, \cdots, k_m))_{m \in \mathbb{N}}$ is a sequence of symmetric piecewise multivariate polynomials with bounded degrees (the degree of $f_t$ does not depend on $m$). Lemma 2 tells us that there exists $m \in \mathbb{N}$, and two vectors $k \in \mathbb{N}^m$ and $k' \in \mathbb{N}^m$ whose coordinates are not equal up to permutations, such that for any $t \in \{0, \cdots, T\}$, $f_t(k_1, \cdots, k_m) = f_t(k'_1, \cdots, k'_m)$. $\quad\square$

**Remark 4.** *Note that in Theorem 3, depth two is minimal: for any pair of non isomorphic rooted trees of depth one, any GNN with one neuron perceptron, an injective activation function, weights set to one, and zero bias can distinguish their root vertex in one iteration. Indeed, in that case, $\xi^1(s) = \sigma(1 + \deg(s))$ if the GNN is recurrent with a combine function given by $\phi : \mathbb{R}^2 \to \mathbb{R}, (x_1, x_2) \mapsto \sigma(x_1 + x_2)$. Hence, $\xi^1(s) \neq \xi^1(s')$ as soon as $\sigma$ is injective and $s$ and $s'$ have distinct degree.*

## 5 ACTIVATIONS THAT ARE NOT PIECEWISE POLYNOMIAL

In this Section we present a proof of Theorem 4. We prove that for any pair of non isomorphic rooted trees of depth two, i.e. trees of the form $T[k_1, \cdots, k_m]$ and $T'[k'_1, \cdots, k'_n]$ (here the $k_i$'s and $k'_i$'s are all greater than or equal to 1, cf. Figure 1) can be distinguished by a bounded GNN with any of the following activation functions: exponential, sigmoid, or a hyperbolic sine, cosine or tangent function. Consider the following 1-neuron perceptron $\phi$ with activation function $\sigma$, $\phi : \mathbb{R}^2 \to \mathbb{R}, \ \phi(x_1, x_2) = \sigma(x_1 + x_2)$. Then it is easy to see that:

$$\forall v \in V(T[k_1, \cdots, k_m]) \quad \xi^1(v) = \sigma(\xi^0(v) + \sum_{w \in N(v)} \xi^0(w)) = \sigma(1 + \deg(v))$$

$$\xi^2(v) = \sigma(\sigma(1 + \deg(v)) + \sum_{w \in N(v)} \sigma(1 + \deg(w)))$$

In particular $\xi^2(s) = \sigma(\sigma(1 + m) + \sum_{i=1}^m \sigma(k_i + 1))$ Now suppose $\sigma$ is either injective on $\mathbb{R}$, or nonnegative and injective on $\mathbb{R}^+$ (this is the case for the exponential, the sigmoid, the hyperbolic tan, and the hyperbolic cosine and sine), $s$ and $s'$ are vertices of two trees with potentially different number of leaves $m$ and $n$, then

$$\xi^2(s) = \xi^2(s') \iff \sum_{i=0}^m \sigma(k_i + 1) = \sum_{i=0}^n \sigma(k'_i + 1) \tag{3}$$

where $k_0 := 1 + m$ and $k'_0 := 1 + n$. The goal of the remainder of this section is to prove that the right hand side equality of Statement 3 implies $m = n$ and $k_i$'s are the same as $k'_i$'s, up to a permutation, for the activation functions $\sigma$ of Theorem 4.

**Theorem 5** (Lindemann-Weierstrass Theorem, 1885). *If $\alpha_1, \cdots, \alpha_n$ are distinct algebraic numbers, then the exponentials $e^{\alpha_1}, \cdots, e^{\alpha_n}$ are linearly independent over the algebraic numbers.*

**Lemma 4.** *Let $n$ and $m$ be positive integers, and $\alpha_1, \cdots, \alpha_n$ and $\alpha'_1, \cdots, \alpha'_m$ be algebraic numbers. Then $\sum_{i=1}^n e^{\alpha_i} = \sum_{i=1}^m e^{\alpha'_i}$ if and only if $m = n$ and the $\alpha_i$'s and $\alpha'_i$'s are equal up to a permutation.*

*Proof.* ($\Longleftarrow$) is clear. For ($\Longrightarrow$), by contradiction suppose that the $\alpha_i$'s and $\alpha'_i$'s are not equal up to a permutation. First, if the $\alpha_i$'s (resp. $\alpha'_i$'s) are not distinct one can group them by their number of occurrences in both sums. Then, we would have a linear dependence with integer coefficients of exponentials of integers. This contradicts Theorem 5 (Linderman-Weirstrass). $\square$

*Proof of Theorem 4.* Without loss of generality, suppose the $k_i$'s and $k'_i$'s are ordered in increasing order. For ease of notation, let $\alpha$ and $\alpha'$ be the vectors defined as $\alpha_i = k_i + 1$ for all $i \in \{1, \cdots, m\}$ and $\alpha'_i = k'_i + 1$ for all $i \in \{1, \cdots, n\}$. We will now prove that Statement 3 implies $\alpha = \alpha'$ in each case.

- $\sigma \in \{\text{sigmoid}, \text{tanh}\}$. In the case of the sigmoid, Statement 3 yields the following equation after multiplication by the the product of the denominators:

$$\left( \sum_{i=1}^m e^{\alpha_i} \left( \prod_{\substack{j=1 \\ j \neq i}}^m (1 + e^{\alpha_j}) \right) \right) \prod_{i=1}^n (1 + e^{\alpha'_i}) = \left( \sum_{i=1}^n e^{\alpha'_i} \left( \prod_{\substack{j=1 \\ j \neq i}}^n (1 + e^{\alpha'_j}) \right) \right) \prod_{i=1}^m (1 + e^{\alpha_i})$$

After developing and grouping each hand side into linear combinations of exponentials we obtain an equation of the form:

$$\sum_{\substack{S \subseteq \{1, \cdots, m\} \\ T \subseteq \{1, \cdots, n\}}} \gamma_{S,T} \exp(\alpha_S + \alpha'_T) = \sum_{\substack{S \subseteq \{1, \cdots, m\} \\ T \subseteq \{1, \cdots, n\}}} \gamma_{S,T} \exp(\alpha'_S + \alpha_T) \tag{4}$$

where for $S \subseteq \{1, \cdots, m\}$, $\alpha_S := \sum_{i \in X} \alpha_i$ (resp. for $T \subseteq \{1, \cdots, n\}$, $\alpha'_T := \sum_{i \in X} \alpha'_i$). Note that $\gamma_{\emptyset, T} = 0$ for all subsets $T \subseteq \{1, \ldots, n\}$. We will prove by induction on $m$ (the size of the vector $\alpha$) that in these conditions, Equation 4 implies that $m = n$ and $\alpha = \alpha'$.

*Base case:* If $\alpha$ has size one and $\alpha'$ has size $n > 0$, then the equation boils down to $\exp(\alpha_1) = \sum_{i=1}^n \exp(\alpha'_i)$ which is true if and only if $n = 1$ and $\alpha_1 = \alpha'_1$ using Lemma 4.

*Induction step:* We suppose the following property true for some nonnegative integer $m$: For any nonnegative integers $\alpha_1, \cdots, \alpha_m$ and $\alpha'_1, \cdots, \alpha'_n$, $\sum_{i=1}^m \sigma(\alpha_i) = \sum_{i=1}^n \sigma(\alpha'_i) \implies m = n$ and $k = k'$.

Suppose now that $\sum_{i=1}^{m+1} \sigma(\alpha_i) = \sum_{i=1}^n \sigma(\alpha'_i)$. Since $\gamma_{\emptyset, T} = 0$ for all $T \subseteq \{1, \ldots, n\}$, the smallest term on the left hand side of equation 4 is $\exp(\alpha_1)$ and the smallest term on the right hand side is $\exp(\alpha'_1)$. Using Lemma 4, this implies that $\alpha_1 = \alpha'_1$. Therefore $\sum_{i=2}^{m+1} \sigma(\alpha_i) = \sum_{i=2}^n \sigma(\alpha'_i)$. We can apply the induction assumption on the vector $(\alpha_2, \cdots, \alpha_{m+1})$ of size $m$ to obtain that $m = n - 1$ and $(\alpha_{k_2}, \cdots, \alpha_{m+1}) = (\alpha_{k'_2}, \cdots, \alpha'_{m+1})$. This proves that $m + 1 = n$ and $\alpha = \alpha'$, which ends the induction.

If $\sigma = \text{tanh} = \frac{\exp(2\cdot) + 1}{\exp(2\cdot) - 1}$. Equation 3 becomes after multiplication by the the product of the denominators:

$$\left( \sum_{i=1}^n (e^{2\alpha_i} - 1) \prod_{j=1, j \neq i}^n (e^{2\alpha_j} + 1) \right) \prod_{j=1}^m (1 + e^{2\alpha'_j}) = \left( \sum_{i=1}^m (e^{2\alpha'_i} - 1) \prod_{j=1, j \neq i}^m (e^{2\alpha'_j} + 1) \right) \prod_{j=1}^n (1 + e^{2\alpha_j})$$

After developing into a linear combination of exponentials on each side, the arguments containing $\alpha_T$ with $T \neq \emptyset$ on the left hand side and $\alpha'_T$ with $T \neq \emptyset$ on the right hand side have positive algebraic coefficients. There are also arguments of the form $\alpha'_T$ on the left hand side and $\alpha_T$ on the right hand side (in other words, $\gamma_{\emptyset, T} \neq 0$, unlike the sigmoid case). However, note that the coefficients corresponding to these terms are (algebraic and) negative. Hence, as a consequence of Lemma 4, the arguments with negative coefficients in front of the exponentials must match up on each side, and we are left with an equation similar to Equation 4 (the arguments have a factor 2), where again $\gamma_{\emptyset, T} = 0$. We can apply the same reasoning by induction as for the sigmoid case, to prove that $\alpha = \alpha'$.

- $\sigma \in \{\sinh, \cosh\}$. If $\sigma = \cosh$, then Equation 3 becomes:

$$\left(\sum_{0=1}^{n} \exp(i\alpha_j) - \sum_{0=1}^{m} \exp(i\alpha_j')\right) + \left(\sum_{j=0}^{m} \exp(-i\alpha_j') - \sum_{0=1}^{n} \exp(-i\alpha_j)\right) = 0$$

Due to Lemma 4, this can only happen if $m = n$ and for all $j \in \{1, \cdots, n\}$, $\alpha_j = \alpha_j'$, because $i\alpha_j, i\alpha_j'$ are algebraic for any $j \in \{1, \cdots, n\}$, and the $\alpha_j$'s and $\alpha_j'$'s are ordered and positive. We conclude that $\alpha = \alpha'$. The case $\sigma \in \{\sinh\}$ can be treated similarly. $\qquad\square$

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
