# OpenReview forum: "On the power of graph neural networks and the role of the activation function"
_ICLR.cc/2024/Conference — ICLR 2024 Conference Withdrawn Submission_

### Official Review · Reviewer_QzRv · 2023-10-29

**Soundness:** 3 good
**Presentation:** 2 fair
**Contribution:** 3 good
**Rating:** 5
**Confidence:** 3

**Summary:**

The paper aims to study the expressive power of GNNs. In particular, the authors prove that for GNNs with piecewise polynomial activations with fixed depth, there exists a pair of rooted trees of depth 2 such that the GNN is not able to distinguish the root of the two trees. The authors further show that if instead one allows for activations that are not piecewise polynomial, then any two root nodes are distinguishable.

**Strengths:**

I believe that the results in the paper are interesting. The result showcasing that restricting activation function to piecewise polynomial functions hinders the expressivity of GNNs so significantly on such a task is a very interesting one. To the best of my knowledge, such a type of result is new, although I am not overly familiar with the existing literature. Studying the role of activation functions is definitely of interest and contributes to a better understanding of non-linear models.

**Weaknesses:**

(W1) The paper is relatively well-written, but is at times hard to follow, especially for someone not working directly in this area. Unfortunately, this might be the nature of such a work, but it would be useful to add more intuition to the manuscript. The work uses specialized mathematics and providing a brief background/intuition would be very beneficial.

For instance, it may be useful to provide a brief background on the algebra of symmetric polynomials and extremal combinatorics. It would also be useful to provide intuition before stating results to help the reader follow the work. These may be added to the main manuscript or in the appendix. In general, I believe that some proofs could be moved to the appendix to make space for more intuition. At the moment I feel like this paper is accessible to a small part of the community and this could help broaden the audience of the paper.

(W2) The statement on trees is interesting, but I am wondering if anything could be said more generally (i.e. what is the role of the activation function for other types of graphs - are piecewise polynomials always unpreferable?). I think a discussion of this sort may add an interesting future direction for further work on the study of non-linearities and bounded-depth GNNs.

**Questions:**

Regarding (W2).

(Q1) How could this work be extended to a discussion beyond the specific tree structures studied in this work?

(Q2) Are piecewise polynomial functions always unpreferable in terms of expressive power?

---

> ### Author Response · Authors · 2023-11-12
>
> Thank you very much for your careful and insightful review.
> Unfortunately, given a comment brought to our attention by one of the reviewers, we have decided to withdraw our article from ICLR to implement significant revisions before resubmitting it for peer review.

---

### Official Review · Reviewer_2Pnx · 2023-10-31

**Soundness:** 3 good
**Presentation:** 3 good
**Contribution:** 2 fair
**Rating:** 1
**Confidence:** 5

**Summary:**

Don't wish to answer

**Strengths:**

Don't wish to answer

**Weaknesses:**

I have reviewed this paper already in a different venue. In the review I wrote
"A very relevant (and recent) reference which is missing is "Exponentially Improving the Complexity of Simulating the Weisfeiler-Lehman Test with Graph Neural Networks" By Aamand et al. https://openreview.net/forum?id=AyGJDpN2eR6 Particularly relevant is the part about lower bounds in that paper. To my understanding this part shows that for ReLU networks at least, bounded size networks cannot separate like 1-WL. The work in this paper is still more general since it addresses piecewise polynomial activations, but it would be good if the authors try to explain why this is significant..."

I was displeased to see that the authors still do not cite this paper, and still claim that they answer the open question by Grohe  "Do bounded GNNs have the same separation power as unbounded GNNs and color refinement?" though this was already done by Aamand et al. I think this behavior is unethical and accordingly set the score of the paper to 1

**Questions:**

Don't wish to answer

**Details Of Ethics Concerns:**

Knowingly not citing relevant work

---

> ### Author Response · Authors · 2023-11-12
>
> We understand the reviewer's comments. We agree that the suggested article in an earlier review provides an answer to the question of [Grohe, 2021] by providing lower bounds on the size of ReLU neural networks needed to express color refinement.
>
> After our previous exchange with the reviewers, we added some changes to our paper. However, in the rush to submit it to ICLR, we forgot to add a presentation and comparison with the work of [Aamand et al.]. We sincerely regret this. We are currently repositioning the ArXiv version of our paper to clarify our main contributions. In particular, we will make it clear that our approach adds to the negative answer (bounded GNNs are weaker than unbounded ones) already provided by the work of [Aamand et al.] by extending it to handle any piecewise polynomial activation functions. We also provide some insights about the case of non piecewise polynomial activation functions, showing that even a single neuron can handle the counterexamples that are problematic for the piecewise polynomial case. We will do our best to contrast and compare with their contribution fairly in terms of the resolution of Grohe's question.
>
> Following this review, we decided to retract the paper in order to bring these changes before re-submitting for peer review.

---

### Official Review · Reviewer_b3We · 2023-11-02

**Soundness:** 2 fair
**Presentation:** 2 fair
**Contribution:** 3 good
**Rating:** 6
**Confidence:** 3

**Summary:**

The manuscript studies the expressive power of Graph neural networks (GNNs). In contrast to the unbounded GNNs (whose size is allowed to change with the graph size) case, the author showed a negative result that GNNs have a lower capability of distinguishing even for tree structures when the activation functions are piecewise polynomials. In addition, they showed that if the activation functions are changed to others that are not piecewise polynomials (e.g., sigmoid, tanh) then a single neuron perception can distinguish the trees (that they have considered in the manuscript), further implying the importance of choosing suitable activation function.

**Strengths:**

1. The problem is of sufficient interest since the expressivity of GNNs has received a lot of attentions, and the problem was open for the bounded GNNs case.
2. The proofs are simple but interesting and the implications of the main results bear a significant importance.

**Weaknesses:**

1. Lack of experimental results: It would be better if the authors could demonstrate via experiments that there exists a case where small (corresponding to bounded) GNNs (e.g., with ReLU) do not perform well, while large (corresponding to unbounded) GNNs perform well.

2. The reviewer believes that the writing could be more polished:
- It would have been nice if the authors could use terminologies that are more used to ML communities.
- It would have been nice if the authors could put a notation/definition paragraph in the beginning part.
- Remark 3: The reviewer had a hard time understanding the words (e.g., here, how do we define the number of regions $r$) before getting into the proof of Lemma 2; however, the proof is located on the later page.

**Questions:**

p-6 line 3: $F_m \rightarrow F_M$?

---

> ### Author Response · Authors · 2023-11-12
>
> Thank you very much for your careful and insightful review.
> Unfortunately, given a comment brought to our attention by one of the reviewers, we have decided to withdraw our article from ICLR to implement significant revisions before resubmitting it for peer review.

---

### Official Review · Reviewer_HnF2 · 2023-11-06

**Soundness:** 3 good
**Presentation:** 2 fair
**Contribution:** 2 fair
**Rating:** 3
**Confidence:** 5

**Summary:**

This paper proves two new theoretical results about GNNs:
- Th 3: for any $I$, with piecewise polynomial activation functions, we can construct 2 trees of depth 2 that are not isomorphic but that cannot be distinguished in less than $I$ iterations.
- Th4: for activation functions like exp or sigmoid, any pair of non-isomorphic trees of depth 2 can be distinguished by a single layer GNN in 2 iterations.
The proof relies on the properties of symmetric polynomials and the Lindemann–Weierstrass theorem in transcendental number theory.

**Strengths:**

The result is a new contribution to the theory of GNNs, but the theoretical limitation of polynomial activation functions for feedforward neural networks has been known for long: Leshno, Moshe, et al. "Multilayer feedforward networks with a nonpolynomial activation function can approximate any function." Neural networks 6.6 (1993): 861-867.

**Weaknesses:**

The result presented in this paper is minimal as it only deals with an example of a tree of depth 2. It is nice to see the possible limitations of GNNs on simple examples, but these theoretical results will probably have a very limited impact in practice.

**Questions:**

You should probably move remark 3 later in the paper. At this stage, this remark cannot be understood as you did not introduce the notion of region yet. Moreover, $r$ is not defined, and you should explicitly give this parameter a name in Definition 1.

In lemma 2, I do not understand why you are considering a sequence $f_p$ of symmetric polynomial functions, as your claim is only for $f_m$. You do not need to define a whole sequence if you consider only one member of this sequence?

In Lemma 3, you must prove that $\phi$ is a piecewise multivariate polynomial. I think the degree of $\phi$ depends on the number of layers in the feedforward neural network defining the combination function of the GNN. As far as I understand, I think this is this degree that should appear in your remark 3.

---

> ### Author Response · Authors · 2023-11-12
>
> Thank you very much for your careful and insightful review.
> Unfortunately, given a comment brought to our attention by one of the reviewers, we have decided to withdraw our article from ICLR to implement significant revisions before resubmitting it for peer review.